# Percentile-Based Reference Values of Umbilical Cord Blood Insulin-like Growth Factor 1 in Japanese Newborns

**DOI:** 10.3390/jcm11071889

**Published:** 2022-03-29

**Authors:** Nobuhiko Nagano, Daichi Katayama, Koichiro Hara, Yuki Sato, Satomi Tanabe, Masako Aoki, Ryoji Aoki, Ichiro Morioka

**Affiliations:** Department of Pediatrics and Child Health, Nihon University School of Medicine, 30-1, Oyaguchi-kamimachi, Itabashi-ku, Tokyo 1738610, Japan; katayama.daichi@nihon-u.ac.jp (D.K.); hara.koichiro@nihon-u.ac.jp (K.H.); sato.yuki@nihon-u.ac.jp (Y.S.); tanabe.satomi@nihon-u.ac.jp (S.T.); aoki.masako@nihon-u.ac.jp (M.A.); aoki.ryoji@nihon-u.ac.jp (R.A.); morioka.ichiro@nihon-u.ac.jp (I.M.)

**Keywords:** birth weight, gestational age, prematurity, small for gestational age, standard deviation score

## Abstract

We aimed to create percentile-based reference values of the umbilical cord blood insulin-like growth factor-1 (IGF-1) levels in Japanese newborns, as these values have not yet been established. A total of 259 newborns were classified into four gestational-age-at-birth (GA) groups: extremely preterm (<28 weeks); early preterm (28–33 weeks); late preterm (34–36 weeks); and term (≥37 weeks). They were further subclassified as small-for-gestational-age (SGA) or non-SGA. The 10th, 25th, 50th, 75th, and 90th percentiles of the umbilical cord blood IGF-1 levels were calculated and compared between the groups by using reference values of 9, 18, 33, 52, and 71 ng/mL, respectively. In the extremely preterm group, the IGF-1 levels were significantly lower than those in the early preterm, late preterm, and term groups (13.5, 24.0, 44.5, and 47.5 ng/mL, respectively; *p* < 0.001). The umbilical cord blood IGF-1 levels in the SGA newborns were significantly lower than those in the non-SGA newborns in all subgroups. In multivariate analyses, the GA and birth weight standard deviation scores were independent determinant factors for the umbilical cord blood IGF-1 levels. Thus, we established percentile-based reference values of umbilical cord blood IGF-1 in Japanese newborns; these reference values can be applied on the basis of the extent of prematurity and the SGA status.

## 1. Introduction

Insulin-like growth factor 1 (IGF-1) is a protein that is encoded by the *IGF1* gene in humans [1,2]. IGF-1 is a single-stranded polypeptide that consists of 70 amino acids, with a molecular weight of 7649; it also has three disulfide bonds in its structure [3]. IGF-1 plays an important role in fetal and neonatal growth, as well as in cardiovascular functions, central nervous system functioning, and lung development [4,5,6]. It has been reported that long-term low serum IGF-1 levels from birth are associated with postnatal growth restriction, brain developmental disorders, retinopathy of prematurity (ROP), and chronic lung disease [7,8,9,10,11].

Previous studies have shown that the value of the IGF-1 at birth is correlated with the birth weight (BW) and the gestational age at birth (GA), and that the IGF-1 values in small-for-gestational-age (SGA) newborns are lower than those in non-SGA newborns [12,13]. The patients in previous reports with regard to the IGF-1 values were term or preterm newborns and term SGA newborns [14,15,16,17]. However, to date, no detailed studies have been conducted where extremely preterm-to-term newborns were included, especially when evaluating the IGF-1 reference values by using umbilical cord blood. In this study, we aimed to create percentile-based reference values for the umbilical cord blood IGF-1 levels in Japanese newborns on the basis of the sex, the GA, and the SGA or non-SGA status.

## 2. Materials and Methods

### 2.1. Study Design

A hospital-based retrospective cohort study was conducted at Nihon University Itabashi Hospital, Tokyo, Japan. This study was approved by the Ethics Committee of the Nihon University School of Medicine (approval no.: RK-190910-3; date: 20 November 2019). Written informed consent was obtained from the parents of all the enrolled newborns. This study was carried out in accordance with the relevant guidelines and regulations. Newborns that were born between 2019 and 2021 were enrolled and were then classified into one of the four GA groups: extremely preterm (<28 weeks); early preterm (28–33 weeks); late preterm (34–36 weeks); and term (≥37 weeks). Thereafter, the newborns were subclassified as either SGA or non-SGA. “SGA” and “non-SGA” were defined as the <10th percentile and the ≥10th percentile, respectively, for the BW, on the basis of Japanese neonatal anthropometric charts at birth for the GA, the sex, and the mother’s history of childbirth [18]. This study included not only singletons, but also twins and triplets. All newborns from whom the umbilical cord blood at birth was collected during the study period were included in this study.

### 2.2. Umbilical Cord Blood IGF-1 Levels

At birth, the umbilicus was double-clamped, and the cord blood was sampled from the umbilical vein. The median (50th percentile), 10th, 25th, 75th, and 90th percentiles of the umbilical cord blood IGF-1 levels were calculated in order to set up the reference values. The umbilical cord blood IGF-1 levels were measured by using the solid-phase radioimmunoassay (RIA) method, as described in a previous study [19,20]. The RIA is an immunological assay that was first developed as a method for measuring the amount of a hormone in the blood by using a radioisotope [21].

### 2.3. Study Methods

The umbilical cord blood IGF-1 levels were compared between the sexes and the four GA groups in all the SGA and non-SGA newborns. In each GA group, the umbilical cord blood IGF-1 levels were compared between the SGA and non-SGA newborns. To clarify the determinant clinical factors for the umbilical cord blood IGF-1 levels, the GA or anthropometric values at birth, such as the BW, the BW standard deviation score (SDS), the birth length (BL), the BL SDS, the birth head circumference, the birth head circumference SDS, and the birth chest circumference, were identified through regression and multivariate analyses.

### 2.4. Statistical Analyses

To compare the umbilical cord blood IGF-1 levels and other clinical factors at birth between the four GA groups, the Kruskal–Wallis test or a chi-squared test were used. The differences between the individual data sets were tested by the all-pairwise test by using the Steel–Dwass method. The correlation between the umbilical cord blood IGF-1 levels and each anthropometric value or GA was analyzed by a regression analysis (the correlation coefficients were calculated) and a multiple logistic regression analysis. Statistical analyses were performed using JMP, Version 14 (SAS Institute Inc., Tokyo, Japan). A *p*-value < 0.05 was considered statistically significant.

## 3. Results

### 3.1. Umbilical Cord Blood IGF-1 Levels

A total of 259 newborns were enrolled, of which 36 were extremely preterm (SGA: 10; non-SGA: 26); 83 were early preterm (SGA: 28; non-SGA: 55); 90 were late preterm (SGA: 21; non-SGA: 69); and 50 were term (SGA: 19; non-SGA: 31). No statistical difference was observed in the umbilical cord blood IGF-1 levels between the sexes (male newborns: 35 [4–121]; female newborns: 32 [4–93]; *p* = 0.60; Figure 1).

The 10th, 25th, 50th, 75th, and 90th percentiles of the umbilical cord blood IGF-1 levels were calculated and compared between the groups by using reference values of 9, 18, 33, 52, and 71 ng/mL, respectively. The umbilical cord blood IGF-1 levels in the extremely preterm group were significantly lower than those of the early preterm, late preterm, and term groups (median values: 13.5, 24.0, 44.5, and 47.5 ng/mL, respectively; *p* < 0.01; Table 1; Figure 2). However, the IGF-1 levels were not significantly different between the late preterm and term groups (*p* = 0.722; Figure 2).

In each GA group, the umbilical cord blood IGF-1 levels were lower in the SGA newborns than in the non-SGA newborns (SGA: 9.5, 10.5, 25.0, and 38.0 ng/mL; non-SGA: 19.0, 38.0, 50.0, and 55.0 ng/mL, respectively; *p* < 0.01; Table 2). In the SGA newborns, the umbilical cord blood IGF-1 values showed no significant difference between the extremely preterm and early preterm newborns (median values: 9.5 and 10.5 ng/mL, respectively; *p* = 0.855; Figure 3). In the non-SGA newborns, the IGF-1 levels were significantly different between the extremely preterm and the early preterm newborns (median values: 19.0 and 38.0 ng/mL, respectively; *p* < 0.001; Figure 4).

### 3.2. Correlation between Umbilical Cord Blood IGF-1 Levels and Gestational Age or Anthropometric Measurements at Birth

The correlations between the umbilical cord blood IGF-1 levels and each anthropometric value or GA are shown in Table 3. All of the anthropometric values (GA, BW, BW SDS, BL, BL SDS, birth head circumference, birth head circumference SDS, birth chest circumference, and IGF-1 levels) showed significantly positive correlations (*p* <0.001; Table 3).

### 3.3. Multivariate Analyses

In the multivariate logistic regression analyses using the GA, BW SDS, BL SDS, and birth head circumference SDS, the GA and the BW SDS were the independent determinant factors for the umbilical cord blood IGF-1 levels (Table 4).

## 4. Discussion

In the present study, we established percentile-based reference values of umbilical cord blood IGF-1 in Japanese newborns. The median umbilical cord blood IGF-1 value was around 50 ng/mL at ≥34 weeks of GA, and it did not change between the late preterm and term newborns. In the non-SGA newborns, the umbilical cord blood IGF-1 levels showed a significant difference between the extremely preterm and early preterm newborns, but not in the SGA newborns. In the SGA newborns, the median umbilical cord blood IGF-1 values were extremely low, at approximately 10 ng/mL at <34 weeks of GA. It was rare for the SGA newborns to have higher IGF-1 levels than the non-SGA newborns. The GA and the BW SDS were the independent determinant factors for the umbilical cord blood IGF-1 levels.

### 4.1. IGF-1 Levels and Sex

In a report of the serum IGF-1 levels in preterm newborns at 33 weeks of corrected GA, the mean IGF-1 levels were 23.1 ng/mL (range: 15.44–39.75 ng/mL), with no difference between the male (23.1 ng/mL) and female infants (23.1 ng/mL) [15]. Consistent with these results, no umbilical cord blood IGF-1 level differences were observed according to sex in this study (Figure 1). Therefore, we created reference values for the umbilical cord blood IGF-1 for the male and female newborns combined.

### 4.2. IGF-1 in the Placenta

IGF-1, IGF-2, and their binding protein (IGFBP) are expressed in the placenta and regulate fetal growth by DNA methylation [22]. DNA methylation is an epigenetic mechanism that forms methylated cytosine-phosphate-guanine (CpG) dinucleotides that are known to suppress gene expression [23]. A study report analyzed the placenta of SGA newborns and found the mRNA and protein levels of IGF-1 to be lower than that of appropriate-for-gestational-age (AGA) newborns; however, no difference was observed between the AGA and the large-for-gestational-age newborns [24]. The CpG methylation of the promoter region of the IGF-1 gene was hypermethylated in the SGA newborns when compared with that of the AGA newborns [24]. In our present study, the SGA newborns had lower umbilical cord blood IGF-1 levels than the non-SGA newborns, and, therefore, the DNA methylation of the IGF-1 gene may be involved.

### 4.3. IGF-1 Levels in SGA Newborns

The production of IGF-1 during the fetal period is regulated by nutritional substances that are supplied by the mother, such as amino acids and glucose [25,26]. Previous reports of cord blood IGF-1 levels are strongly associated with the birth weight, the leptin level, the fat mass, and the body fat percentage, which indicate that IGF-1 is an important factor in the fetal fat accumulation in the uterus [27]. Another previous study reports that newborns who are born to mothers with pregnancy hypertension had lower cord blood IGF-1 levels compared with those born to mothers without pregnancy hypertension, and the SGA newborns showed low cord blood IGF-1 levels [28]. Consistent with the results, in our present study, the cord blood IGF-1 levels were lower in the SGA newborns compared to those of the non-SGA newborns in all four GA groups. The umbilical cord blood IGF-1 levels in the preterm SGA newborns who were born at 28–33 weeks of GA were as low as the extremely preterm newborns who were born at <28 weeks of GA. SGA was also an important determinant factor for the cord blood IGF-1 levels. In the SGA newborns, the umbilical cord blood IGF-1 levels were remarkably low in extremely preterm and early preterm newborns who were born at <34 weeks of GA, when compared with those of late preterm newborns who were born at 34–36 weeks of GA, and term newborns who were born at ≥37 weeks of GA. We previously reported that preterm SGA newborns had higher rates of short stature at three years of life than late preterm or term SGA newborns [29]. A reason for this difference in the incidence of short stature in early childhood may be related to the blood IGF-1 levels at birth.

### 4.4. IGF-1 Levels in Extremely Preterm Newborns

In our present study, in the extremely preterm newborns, the median IGF-1 values were as low as 13.5 ng/mL, which may indicate a low nutrition status. A clinical study in 87 newborns born between 24 and 32 weeks of GA evaluated the relationship between the serum IGF-1 levels and the nutrition up to 36 weeks of corrected GA, and it reveals that the parenteral nutrition and the IGF-1 levels at 2 weeks of age were inversely proportional [14]. The total energy intake was positively associated with the serum IGF-1 levels, especially at 30–33 weeks of corrected GA [14]. After birth, extremely preterm newborns lose nutrition from their mother and become dependent on parenteral nutrition for several weeks, which may result in low levels of IGF-1 for a prolonged period.

### 4.5. Association between IGF-1 Levels and Neonatal Diseases

In a study of preterm newborns that weighed under 1251 g in the United States, the mean IGF-1 value at 28–33 weeks of corrected GA without ROP was 20.0 ng/mL, and, at Stage 1 or 2, the ROP was 18.0 ng/mL, and, at Stage 3, the ROP was 17.0 ng/mL, which suggests that the IGF-1 values were associated with the ROP development and severity [30]. Therefore, newborns with low serum IGF-1 levels after birth may be predisposed to ROP and may require treatment. It has also been reported that low serum IGF-1 levels were strongly associated with various neonatal complications in animal models (periventricular leukomalacia model rat; cerebral hypoxic ischemia model rat; primary IGF-1-deficient mice; and a necrotizing enterocolitis model) [31,32,33,34] and humans (poor weight gain and brain weight, and the development of ROP and bronchopulmonary dysplasia) [7,9,10,11]. Early IGF-1 supplementation, such as the intravenous supplementation of human recombinant IGF-1 to attain normal serum levels, may reduce these complications and promote growth and development in extremely preterm newborns with/without SGA. Clinical trials are currently underway to see if early IGF-1 supplementation can improve and reduce preterm newborn morbidity [35,36]. A comparison of our cord blood IGF-1 reference values may be useful for the prediction of complications in extremely preterm newborns. The IGF-1 reference values change slightly with the various measurement methods that are available [19,20]. Therefore, it is important to carefully consider the measurement method when using the reference values. In the future, we expect to obtain the therapeutic effect of IGF-1 supplementation in extremely preterm newborns with low cord blood IGF-1 levels.

### 4.6. Limitations

This study has some limitations. First, this study was conducted with a small cohort in a single Japanese center. Second, this study included only Japanese newborns. It will be necessary to clarify the difference in the IGF-1 values between the different races. Third, we did not investigate the clinical information, such as neonatal diseases, the nutritional status, the insulin levels, the body weight gain, the smoking of the mother, and gestational diabetes mellitus. Finally, the postnatal serum IGF-1 levels were not measured. However, our results supply evidence for the newborn’s umbilical cord IGF-1 reference values, not only in all newborns, but also in newborns, classified by sex, GA, and SGA or non-SGA status.

## 5. Conclusions

We established percentile-based reference values of umbilical cord blood IGF-1 levels in Japanese newborns. These reference values can be applied on the basis of the extent of the gestational age or the SGA status. A comparison of our cord blood IGF-1 reference values may be useful for predicting neonatal diseases in preterm newborns.

## Figures and Tables

**Figure 1 jcm-11-01889-f001:**
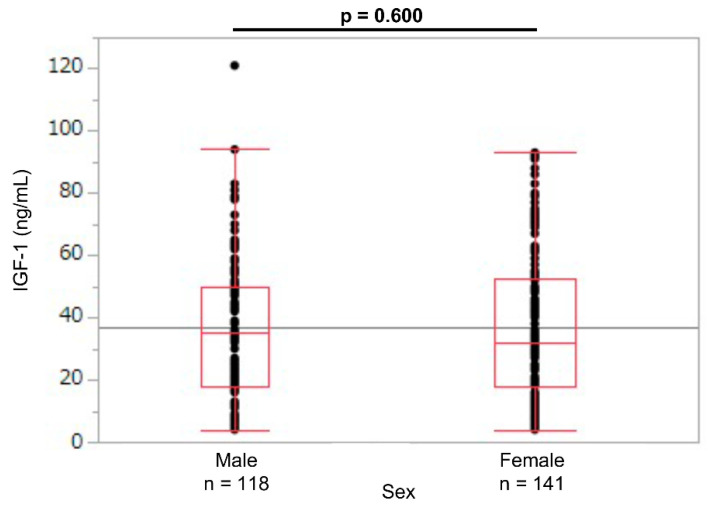
Umbilical cord IGF-1 levels in male and female newborns. IGF-1: insulin-like growth factor 1.

**Figure 2 jcm-11-01889-f002:**
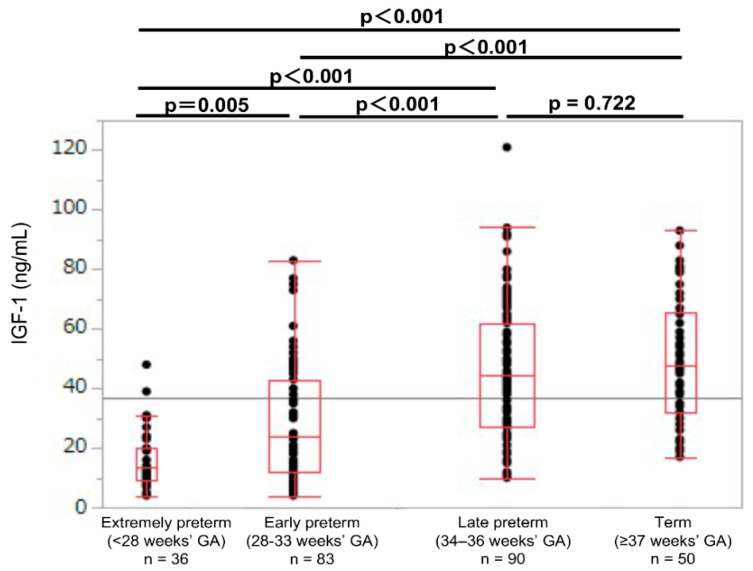
Umbilical cord IGF-1 levels in extremely preterm, early preterm, late preterm, and term newborns. GA: gestational age at birth; IGF-1: insulin-like growth factor 1.

**Figure 3 jcm-11-01889-f003:**
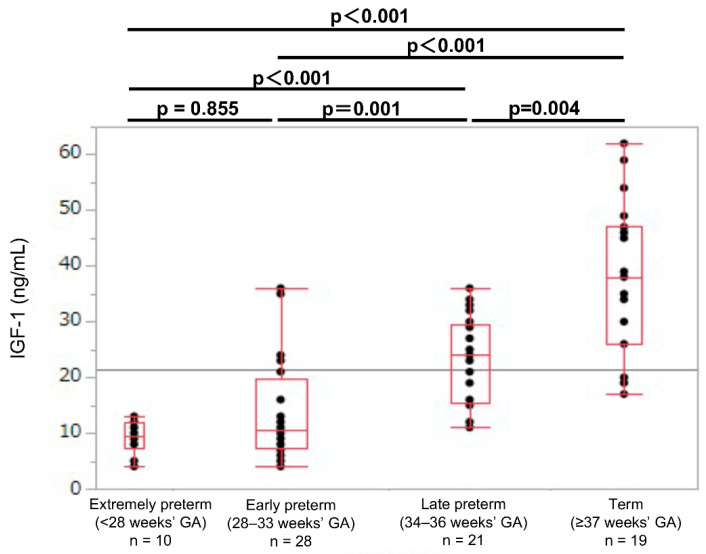
Umbilical cord IGF-1 levels in extremely preterm, early preterm, late preterm, and term SGA newborns. GA: gestational age at birth; IGF-1: insulin-like growth factor 1; SGA: small-for-gestational-age.

**Figure 4 jcm-11-01889-f004:**
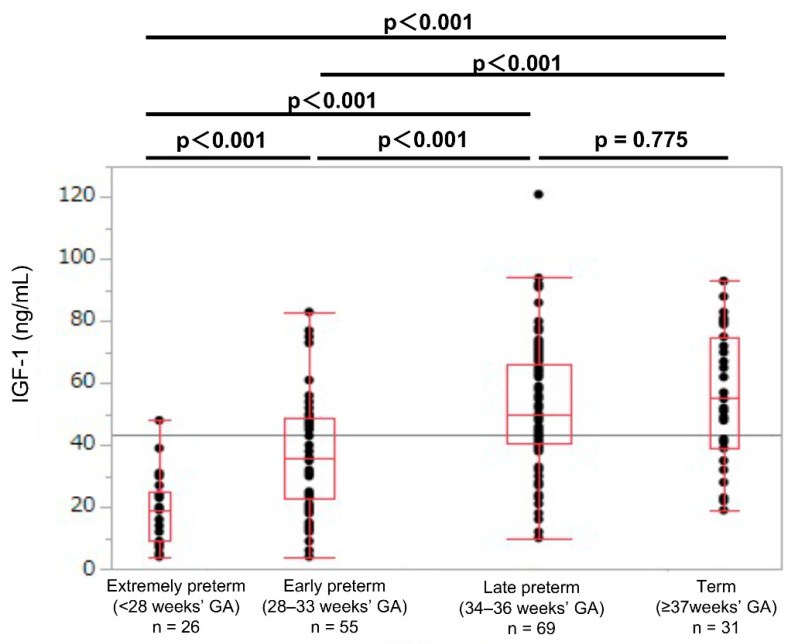
Umbilical cord IGF-1 levels in extremely preterm, early preterm, late preterm, and term non-SGA newborns. GA: gestational age at birth; IGF-1: insulin-like growth factor 1; SGA: small-for-gestational-age.

**Table 1 jcm-11-01889-t001:** Umbilical cord blood IGF-1 levels classified by GA.

	All Newborns *n* = 259	Extremely Preterm (<28 Weeks GA) *n* = 36	Early Preterm (28–33 Weeks GA) *n* = 83	Late Preterm (34–36 Weeks GA) *n* = 90	Term (≥37 Weeks GA) *n* = 50
GA (weeks)	34 (22–38)	25 (22–27)	31 (28–33)	35 (34–36)	37 (37–38)
BW (g)	1880 (278–3560)	678 (278–1224)	1389 (424–2448)	2255 (1492–3126)	2434 (1744–3560)
BW SDS	−0.54 (−5.0–2.41)	−0.47 (−3.95–1.76)	−0.74 (−5.0–1.69)	−0.38 (−2.96–2.41)	−0.75(−3.0–2.23)
BL (cm)	42.5 (24.0–53.0)	30.5 (24–37.5)	38.7 (27.0–46.0)	44.7 (39.0–51.5)	46.2 (40.0–53.0)
BL SDS	−0.58 (−4.2–2.7)	−0.87 (−3.12–2.7)	−0.73 (−4.2–1.81)	−0.42 (−2.89– +2.3)	−0.62 (−2.92–2.7)
Birth head circumference (cm)	30.5 (17.0–36.0)	22.5 (17.0–27.0)	28.0 (22.3–33.5)	31.6 (20.0–35.5)	32.9 (30.0–36.0)
Birth head circumference SDS	−0.1 (−2.72–2.51)	−0.175 (−2.32–2.37)	−0.1 (−2.72–1.92)	−0.06 (−1.95–2.51)	−0.07 (−2.04–2.3)
Birth chest circumference (cm)	27.0 (15.5–35.0)	18.6 (15.5–22.6)	24.4 (15.5–29.5)	28.6 (24.5–35.0)	30.0 (25.0–33.5)
Male	118 (46)	13 (41)	38 (46)	39 (43)	25 (50)
SGA	78 (30)	8 (25)	28 (34)	21 (23)	19 (38)
IGF-1 (ng/mL) (10%ile)	9.0	5.0	7.0	16.2	20.2
IGF-1 (ng/mL) (25%ile)	18.0	9.0	12.0	27.0	32.0
IGF-1 (ng/mL) (50%ile)	33.0 (4–121)	13.5 (4–48)	24.0 (4–83)	44.5 (10–121)	47.5 (17–93)
IGF-1 (ng/mL) (75%ile)	52.0	20.0	43.0	62.0	65.5
IGF-1 (ng/mL) (90%ile)	71.0	30.7	54.0	73.9	79.9

Data are shown as median (range) or number (percentage). BL: birth length; BW: birth weight; GA: gestational age at birth; IGF-1: insulin-like growth factor 1; SDS: standard deviation score; SGA: small-for-gestational-age.

**Table 2 jcm-11-01889-t002:** Comparison of umbilical cord blood IGF-1 levels between SGA and non-SGA newborns.

**Extremely Preterm**			
	**Extremely Preterm (<28 Weeks GA)**		***p*-Value**
	**SGA (*n* = 10)**	**Non-SGA (*n* = 26)**	
GA (weeks)	26 (22–27)	25 (22–27)	0.037
BW (g)	551 (278–690)	770.5 (562–1224)	<0.001
BW SDS	−2.6 (−3.95–−1.7)	−0.25 (−1.25–1.76)	<0.001
BL (cm)	29.0 (24.0–35.5)	31.75 (26.5–37.5)	0.061
BL SDS	−1.78 (−3.12–−0.2)	−0.24 (−2.3–2.7)	0.002
Birth head circumference (cm)	22.1 (17.0–24.0)	23.25 (20.0–27.0)	0.034
Birth head circumference SDS	−1.31 (−2.32–−0.3)	0.335 (−1.4–2.37)	<0.001
Birth chest circumference (cm)	16.7 (15.5–19.0)	19.5 (16.8–22.6)	<0.001
Male	3 (30)	12 (46)	1.000
IGF-1 (ng/mL) (10%ile)	4.1	6.4	
IGF-1 (ng/mL) (25%ile)	7.3	9.0	
IGF-1 (ng/mL) (50%ile)	9.5 (4–13)	19.0 (4–48)	0.007
IGF-1 (ng/mL) (75%ile)	12.0	24.8	
IGF-1 (ng/mL) (90%ile)	12.9	33.4	
**Early Preterm**			
	**Early Preterm (28–33 Weeks GA)**		***p*-Value**
	**SGA (*n* = 28)**	**Non-SGA (*n* = 55)**	
GA (weeks)	31 (28–33)	31 (28–33)	0.646
BW (g)	985.5 (424–1484)	1540 (1016–2448)	<0.001
BW SDS	−2.38 (−5.0–−1.3)	−0.06 (−1.27–1.69)	<0.001
BL (cm)	35.5 (27.0–41.7)	40.4 (35.0–46.0)	<0.001
BL SDS	−2.16 (−4.2–−0.14)	−0.31 (−2.29–1.81)	<0.001
Birth head circumference (cm)	26.6 (22.3–30.0)	29.0 (24.5–33.5)	<0.001
Birth head circumference SDS	−0.815 (−2.48–0.5)	0.31 (−2.72–1.92)	<0.001
Birth chest circumference (cm)	21.0 (15.5–24.5)	25.7 (20.8–29.5)	<0.001
Male	16 (57)	22 (40)	0.138
IGF-1 (ng/mL) (10%ile)	5.0	14.6	
IGF-1 (ng/mL) (25%ile)	7.3	23.0	
IGF-1 (ng/mL) (50%ile)	10.5 (4–36)	38.0 (4–83)	<0.001
IGF-1 (ng/mL) (75%ile)	19.8	49.0	
IGF-1 (ng/mL) (90%ile)	25.1	61.0	
**Late Preterm**			
	**Late Preterm (34–36 Weeks GA)**		***p*-Value**
	**SGA (*n* = 21)**	**non-SGA (*n* = 69)**	
GA (weeks)	35 (34–36)	35 (34–36)	0.327
BW (g)	1816 (1492–2276)	2304 (1701–3126)	<0.001
BW SDS	−1.65 (−2.96–−1.29)	0.0 (−1.27– 2.41)	<0.001
BL (cm)	42.5 (39.0–48.0)	45.5 (40.5–51.5)	<0.001
BL SDS	−1.4 (−2.89–1.32)	−0.07 (−1.53–2.3)	<0.001
Birth head circumference (cm)	30.3 (20.0–33.2)	32.0 (29.0–35.5)	<0.001
Birth head circumference SDS	−0.76 (−1.9–0.46)	0.1(−1.95–2.51)	<0.001
Birth chest circumference (cm)	26.5 (24.5–30.5)	29.0 (25.3–35.0)	<0.001
Male	11 (52)	28 (41)	0.913
IGF-1 (ng/mL) (10%ile)	12.0	23.0	
IGF-1 (ng/mL) (25%ile)	16.0	40.5	
IGF-1 (ng/mL) (50%ile)	25.0 (11–36)	50.0 (10–121)	<0.001
IGF-1 (ng/mL) (75%ile)	29.5	66.0	
IGF-1 (ng/mL) (90%ile)	33.8	78.0	
**Term**			
	**Term (≥ 37 Weeks GA)**		***p*-Value**
	**SGA (*n* = 19)**	**Non-SGA (*n* = 31)**	
GA (weeks)	37 (37–38)	37 (37–38)	0.705
BW (g)	2125 (1744–2532)	2808 (2310–3560)	<0.001
BW SDS	−1.91 (−3.0–−1.31)	−0.01 (−1.27– 2.23)	<0.001
BL (cm)	44.0 (40.0–47.0)	47.3 (43.4–53.0)	<0.001
BL SDS	−1.62 (−2.92–−0.16)	−0.17 (−1.66–2.7)	<0.001
Birth head circumference (cm)	31.7 (30.0–33.5)	34.0 (31.0–36.0)	<0.001
Birth head circumference SDS	−0.76 (−2.04–0.31)	0.69(−1.17–2.3)	<0.001
Birth chest circumference (cm)	28.0 (25.0–31.0)	31.0 (27.5–33.5)	<0.001
Male	11 (58)	14 (45)	0.561
IGF-1 (ng/mL) (10%ile)	19.0	24.0	
IGF-1 (ng/mL) (25%ile)	26.0	39.0	
IGF-1 (ng/mL) (50%ile)	38.0 (17–62)	55.0 (19–93)	0.002
IGF-1 (ng/mL) (75%ile)	47.0	75.0	
IGF-1 (ng/mL) (90%ile)	59.0	82.6	

Data are shown as median (range) or number (percentage). BL: birth length; BW: birth weight; GA: gestational age at birth; IGF-1: insulin-like growth factor 1; SDS: standard deviation score; SGA: small-for-gestational-age.

**Table 3 jcm-11-01889-t003:** Association between insulin-like growth factor-1 levels and gestational age or anthropometric values at birth.

	Correlation Coefficient (95% CI)	*p*-Value
GA	0.518 (0.423–0.602)	<0.001
BW	0.708 (0.642–0.764)	<0.001
BW SDS	0.564 (0.474–0.641)	<0.001
BL	0.609 (0.526–0.681)	<0.001
BL SDS	0.356 (0.244–0.458)	<0.001
Birth head circumference	0.618 (0.536–0.688)	<0.001
Birth head circumference SDS	0.416 (0.310–0.512)	<0.001
Birth chest circumference	0.672 (0.600–0.734)	<0.001

BL: birth length; BW: birth weight; CI: confidence interval; GA: gestational age at birth; SDS: standard deviation score.

**Table 4 jcm-11-01889-t004:** Multivariate analysis.

Factor	Logarithmic Worth (95% CI)	*p*-Value
GA	22.37 (2.17–3.13)	<0.001
BW SDS	13.44 (7.73–12.76)	<0.001
BL SDS	1.00 (−4.12–0.36)	0.100
Birth head circumference SDS	0.09 (−2.49–3.13)	0.822

BL: birth length; BW: birth weight; CI: confidence interval; GA: gestational age at birth; SDS: standard deviation score.

## Data Availability

The data that support the findings of this study are available from the corresponding author upon reasonable request.

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
