# Peer review of "Percentile-Based Reference Values of Umbilical Cord Blood Insulin-like Growth Factor 1 in Japanese Newborns"

_jcm, 2022, doi:10.3390/jcm11071889_

Round 1

Reviewer 1 Report

Some parts has o be corrected:

  1. at material an method is described as retrospective the study.When was performed the cord blood sample taken?
  2. Method used for IGF1 determination in the current study has to be described, not as citation of other older studies
  3. Has to be describe the method used to set up the reference values of each group
  4. replace hypertension disorders of pregnancy with pregnancy hypertension , as is used in all papers, and reformulate the whole paragraph.Formulation is unclear , you are using extremely low value and extremely preterm new born..it creates confusion
  5. At pg 9 you are saying that term newborns over 34 weeks.Term newborns are over 37 weeks.Has to be reformulated
  6. In "limitaition" paragraph reformulate the last phrase
  7.  

Author Response

Response to Reviewer #1:

We would like to thank the reviewer for the thoughtful review and informative comments.

1. at material an method is described as retrospective the study.When was performed the cord blood sample taken?

Response: Umbilical cord blood samples were collected at birth in all cases. It is described with a yellow highlight. Please see in Section 2.2 on page 2.

2. Method used for IGF1 determination in the current study has to be described, not as citation of other older studies

Response: The IGF-1 measurement method used at our hospital was the RIA method, therefore, we described the brief method in Section 2.2 on page 2.

RIA is an immunological assay that was first developed as a method for measuring the amount of a hormone in blood using a radioisotope [21].”

3. Has to be describe the method used to set up the reference values of each group

Response: We added the phrase in Section 2.2 on page 2.

Median (50th percentile), 10th, 25th, 75th, and 90th percentiles of umbilical cord blood IGF-1 levels were calculated to set up the reference values.

4. replace hypertension disorders of pregnancy with pregnancy hypertension , as is used in all papers, and reformulate the whole paragraph. Formulation is unclear , you are using extremely low value and extremely preterm new born..it creates confusion

Response: We changed the words in Section 4.3. on page 8.

Another previous study has reported that newborns born to mothers with pregnancy hypertension had lower cord blood IGF-1 levels compared with those born to mothers without pregnancy hypertension, and SGA newborns showed low cord blood IGF-1 levels [27].

In SGA newborns, the umbilical cord blood IGF-1 levels were remarkably low in extremely preterm and early preterm newborns born at <34 weeks of GA when compared with those of late preterm (at 34-36 weeks of GA) and term newborns born at ≥37 weeks of GA.”

5. At pg 9 you are saying that term newborns over 34 weeks. Term newborns are over 37 weeks. Has to be reformulated

Response: We added and changed the words clearly in Section 4.3. on page 9.

“In SGA newborns, the umbilical cord blood IGF-1 levels were remarkably low in extremely preterm and early preterm newborns born at <34 weeks of GA when compared with those of late preterm at 34-36 weeks of GA and term newborns born at ≥37 weeks of GA.”

6. In "limitation" paragraph reformulate the last phrase You were very clear describing the exclusion criteria and they were very important. Maybe in your discussion or conclusion you should include some comments about ethnical features.

Response: We added the sentence in discussion section.

this study was included only Japanese newborns. It should be necessary to clarify the difference of IGF-1 values between the different races.

Again, we thank you for your useful comments on our paper. We hope that the revised manuscript is suitable for publication.

Reviewer 2 Report

This is a retrospective single center study conducted to provide reference values of umbilical cord blood Insulin-like growth factor-1 in Japanese newborns.The topic is very important and the manuscript is well written, but there are some concerns.

Major points

  1. Inclusion criteria and recruitment details were not clearly stated in the METHODS section.
  2. It has been well known to be the relationship between IGF-1 levels and ROP. However, the author did not analyze the correlation between IGF-1 levels and ROP. Since the authors investigated the IGF-1 levels in extremely preterm birth, it is better to show the results of the relationship between IGF-1 levels and ROP in extremely preterm birth newborns.
  3. Limitation and Discussion. The author should show the details of the relationship between IGF-1 and maternal complication. IGF-1 was known to be associated with maternal insulin, maternal body weight gain, maternal smoking, GDM and maternal nutrition status.

Minor points

  1. Method: This study included twin newborns?
  2. Limitation: This study is single center study.

Author Response

Response to Reviewer #2:

We thank the reviewer for their thoughtful review and insightful comments.

This is a retrospective single center study conducted to provide reference values of umbilical cord blood Insulin-like growth factor-1 in Japanese newborns. The topic is very important and the manuscript is well written, but there are some concerns.

Response: Thank you for your comments.

Major points

  1. Inclusion criteria and recruitment details were not clearly stated in the METHODS section.

Response: We added the sentence in Section 2.1. on page 2.

All newborns of which umbilical cord blood at birth was collected during the study period were included in this study.

  1. It has been well known to be the relationship between IGF-1 levels and ROP. However, the author did not analyze the correlation between IGF-1 levels and ROP. Since the authors investigated the IGF-1 levels in extremely preterm birth, it is better to show the results of the relationship between IGF-1 levels and ROP in extremely preterm birth newborns.

Response: We agree that the association between IGF-1 and ROP is well known. We have analyzed the association between IGF-1 value and ROP and obtained the result that extremely preterm infants with low cord blood IGF-1 levels were at increased risk of ROP. Because the study results are currently submitting to another journal, this present study focused on the reference value of umbilical cord blood IGF-1.

  1. Limitation and Discussion. The author should show the details of the relationship between IGF-1 and maternal complication. IGF-1 was known to be associated with maternal insulin, maternal body weight gain, maternal smoking, GDM and maternal nutrition status.

Response: We agree with the reviewer. However, unfortunately, we did not consider the maternal complications in this study. We added the sentence as a limitation.

“Third, we did not investigate the clinical information, such as neonatal diseases, the nutritional status, insulin levels, body weight gain, and smoking of the mother, and gestational diabetes mellitus.”

Minor points

  1. Method: This study included twin newborns?

Response: Yes. We added the sentence in Section 2.1. on page 2.

This study was included not only singleton, but also twins and triplets.

  1. Limitation: This study is single center study.

Response: We have already described the following sentence with yellow highlight in the limitation section.

“This study has some limitations. First, this study was conducted with a small cohort in a single Japanese center.”

Again, we would like to thank you for your valuable comments on our paper. We hope that the revised manuscript is suitable for publication.
